# Effect of Maintenance and Water–Cement Ratio on Foamed Concrete Shrinkage Cracking

**DOI:** 10.3390/polym14132703

**Published:** 2022-07-01

**Authors:** Chunbao Li, Xiaotian Li, Shen Li, Di Guan, Chang Xiao, Yanyan Xu, Valentina Y. Soloveva, Hojiboev Dalerjon, Pengju Qin, Xiaohui Liu

**Affiliations:** 1College of Pipeline and Civil Engineering, China University of Petroleum (East China), Qingdao 266580, China; z21060136@s.upc.edu.cn (X.L.); z21060135@s.upc.edu.cn (C.X.); 2Construction Project Management Branch of China National Petroleum Pipeline Network Group Co., Ltd., Langfang 065001, China; lishen@pipechina.com.cn (S.L.); guandi@pipechina.com.cn (D.G.); 3Henan Huatai New Material Technology Corp., Ltd., Nanyang 473000, China; jennifer@gmail.com; 4Emperor Alexander I ST Petersburg State Transport University, St. Petersburg 190031, Russia; soloviova-pgups@mail.ru; 5Mining-Metallurgical Institute of Tajikistan, Buston City 735730, Tajikistan; gmit_tajikistan@mail.ru; 6College of Civil Engineering, Taiyuan University of Technology, Taiyuan 030024, China; qinpengju@tyut.edu.cn; 7Qingdao Urban Development Group Co., Ltd., Qingdao 266061, China; liuxiaohui2014upc@163.com

**Keywords:** foamed concrete, plastic shrinkage and cracking, concrete curing, water–cement ratio

## Abstract

This is a study on how to reduce shrinkage and improve crack resistance of foamed concrete. By selecting different curing temperatures and humidity, six different curing conditions were analyzed. The shrinkage deformation and maximum crack width of foamed concrete blocks with water–cement ratios of 0.4 and 0.5, under six curing conditions, were measured by a comparator and optical microscope, and the cracking time was recorded. The effects of curing temperature, humidity and water–cement ratio on the shrinkage and crack resistance of the foamed concrete were analyzed by comparing the experimental results of each group. We studied the primary and secondary order of the three factors affecting the drying shrinkage of foamed concrete. The results show that: temperature is the primary factor that changes the drying shrinkage performance of foamed concrete, followed by the water–cement ratio, and finally humidity. The interaction of these three factors is not obvious. The shrinkage of foamed concrete increases with the increase in temperature; increasing the humidity of curing can control the water loss rate of foamed concrete and reduce shrinkage. Lower humidity and higher temperature will make cracks appear earlier; with an increase in the water–cement ratio, the initial cracking time is shortened and the cracking property of foamed concrete is improved.

## 1. Introduction

Traditional foamed concrete is defined as a lightweight, porous concrete material, which is made by adding foam into slurry consisting of siliceous and calcareous components, water and admixture, followed by proper curing in certain conditions. The volume density of foamed concrete is 100–1600 kg/m^3^, the thermal conductivity is usually 0.05~0.46 W/(m·K), and the compressive strength is 0.2–30 MPa. The industry standard *“Technical Specification for Application of Foamed Concrete”* (JGJ/T 341-2014) [1] defines foamed concrete as a concrete where cement is the main cementitious material and bubbles are added to the slurry made of aggregate, admixture and water. The lightweight porous concrete with a closed cell structure is formed after mixing, stirring, pouring, molding, and maintenance.

In recent years, China has continuously implemented reform of thermal insulation materials and building energy conservation policies, putting forward new requirements for thermal insulation materials and energy-saving materials, such as light weight, thermal insulation and high strength [2]. Foamed concrete is an ideal new material for heat preservation and energy conservation. It has comprehensive advantages over traditional materials, such as being lightweight, having sound insulation, fire resistance, being waterproof, having heat preservation and environmental protection [3]. Foamed concrete is widely used in tunnel, highway and construction projects [4]. However, in practical applications, due to the influence of the curing system, water–cement ratio and raw material properties [5,6,7,8], foamed concrete is exposed to a series of problems, such as low strength and large amounts of dry shrinkage. How to reduce the shrinkage of foamed concrete and improve its crack resistance has become an important subject in foamed concrete engineering technology.

The China–Russia Eastern trunk line starts from the China–Russia border in Heihe City, Heilongjiang Province, runs through Heilongjiang, Jilin, Inner Mongolia, Liaoning and Heilongjiang, Beijing, Tianjin, Shandong, Jiangsu, Shanghai, and ends at Baihe End Station in Shanghai, with a total length of 3334.6 km [9]. The design output is 380 × 108 m^3^/a, the design pressure is 12 MPa/10 MPa, and the pipe diameter is 1422 mm/1219 mm/101.6 mm. According to the overall direction of the route, the China–Russia East Route pipeline crosses the Yangtze River in Nantong Economic and Technological Development Zone and Changshu Economic and Technological Development Zone in Jiangsu Province, via the shield tunnel, which is the control project of the China–Russia East Route. The Yangtze River shield tunnel is huge, with a horizontal length of 10,324 m. The horizontal length of the shield tunnel crossing is 10226 m (north bank shaft center to south bank shaft center), as shown in Figure 1. According to the preliminary design scheme of the project, there are plans to fill the tunnel with foamed concrete after the installation of pipes, as shown in Figure 2.

Domestic and foreign scholars have conducted much research on foamed concrete. Deng J. et al. studied the influence of different lengths and different dosages of polyvinyl alcohol fibers on the water absorption, strength and shrinkage of foamed concrete [10]. Some scholars have studied the influence of mix ratio on concrete [11,12]. Rong H. et al. used yeast as a raw material to prepare microbial foaming agents and studied the influence of foaming technology on the performance and microstructure of foamed concrete [13]. Mohamad Noridah et al. studied the effect of fiber on the properties of foamed concrete [14,15]. Othman Rokiah. et al. studied the relationship between foamed concrete density and compressive strength [16]. Abdullah Faisal Alshalif et al. studied the effect of density and curing conditions on CO_2_ isolation in foamed concrete brick, and the effect of bacterial self-healing and CO_2_ bio-sealing on the strength of bio foamed concrete brick [17,18]. Yang S. et al. studied the preparation and properties of low-density foamed concrete [19]. Hou Li et al. studied the influence of foaming agent on foamed concrete [20]. Chen Y. et al. studied the application of foamed concrete in backfilling [21].

Combined with research from scholars at home and abroad, it has been found that, in engineering applications, foamed concrete has many advantages when compared with traditional concrete; however, there is a lack of systematic research on the factors affecting the shrinkage and cracking of foamed concrete. In this paper, typical working conditions were simulated within different scenarios, and the shrinkage and maximum crack value of foamed concrete in these conditions were studied. This paper aims to systematically summarize the influence of curing system and water–cement ratio on the cracking and shrinkage of foamed concrete, and to provide theoretical guidance for the filling and construction of foamed concrete.

At present, the problem of controlling the shrinkage cracking of foamed concrete is mostly seen in the research of additives, material ratio, fibers, etc. Based on this, this study formulates six curing systems, selects two water–cement ratios, and obtains the shrinkage and cracking performance of foamed concrete under different curing conditions and water–cement ratios, in order to systematically summarize the effects of temperature, humidity and water–cement ratio on the shrinkage and cracking of foamed concrete, and provide theoretical guidance for the filling and construction of foamed concrete.

## 2. Materials and Methods

### 2.1. Materials and Specimen Preparation

#### 2.1.1. Cement

P.O 42.5 grade ordinary Portland cement was produced by Huaxin Cement (Nantong, China) Co., Ltd., and the physical properties of the cement are shown in Table 1.

#### 2.1.2. F2.1.2 Fly Ash

The fly ash, grade I, was sourced from Nanjing Huaneng Power Plant (Nanjing, China). The chemical composition of fly ash is shown in Table 2.

#### 2.1.3. Foaming Agent

Foaming agent for HT foamed concrete is present in Figure 3.

#### 2.1.4. Water

Tap water is used, and, according to foamed concrete density grade A11, the compositions in 1 m^3^ foamed concrete are shown in Table 3.

#### 2.1.5. Admixtures

HTFC foamed concrete admixture is provided by Henan Huatai New Material Technology Co., LTD. (Henan, China).

#### 2.1.6. Preparation of Foamed Concrete

The preparation of foamed concrete includes foam preparation and cement mortar preparation and casting [22]. Foam was prepared by adding foaming agent into a foaming machine. Fly ash, admixtures and cement slurry were stirred evenly with a blender at a low speed for about 2 min, followed by adding water to prepare a homogenous slurry. Foam made by the foaming equipment was then added into the slurry. The mixed slurry was cast into a prepared mold, and a spatula was used to smooth the surface.

### 2.2. Design of Experiments

#### 2.2.1. Designed Maintenance Regime

Given the possible environments that foamed concrete may encounter in this project, six typical environmental conditions, including temperature and humidity, were selected to simulate the actual conditions of the project: (1) temperature 25 ± 1 °C and relative humidity 75 ± 5%; (2) temperature 25 ± 1 °C and relative humidity 15 ± 5%; (3) temperature 45 ± 1 °C and relative humidity 75 ± 5%; (4) temperature 45 ± 1 °C and relative humidity 15 ± 5%; (5) temperature 65 ± 1 °C and relative humidity 75 ± 5%; (6) temperature 65 ± 1 °C and relative humidity 15 ± 5%.

#### 2.2.2. Drying Shrinkage Test

The drying shrinkage of each specimen was measured 1 day after demolding, and continued for 28 days. The detailed test procedure was as follows:
(1)Steel molds measuring 40 × 40 × 160 mm were cleaned and lubricated for easier demolding. In addition, due to the high fluidity of foamed concrete, a layer of petroleum jelly was coated around each test mold to prevent the slurry from seeping out.(2)The foamed concrete slurry was cast in steel molds at the same temperature and humidity as stated in the designed curing regime.(3)The specimens were demolded after 24 h and placed in a curing box, with constant temperature and humidity control. The shrinkage of each specimen was measured after curing for different times, according to the test scheme. The test instrument and specimen are shown in Figure 4.(4)The drying shrinkage strain is determined as follows:(1)$$\varepsilon_{st}=\frac{L_{0}-L_{t}}{L_{t}}\times 100\%$$


$\varepsilon_{st}$—shrinkage strain of concrete at age *t*, where *t* is the initial measurement time of drying shrinkage;

$L_{0}$—initial length of prismatic specimen (mm);

$L_{t}$—the length of a prismatic specimen at time *t* (mm).

#### 2.2.3. Early Cracking

(1)The mixed foamed concrete was cast into an elliptical mold (inner size 210 mm × 90 mm × 45 mm, outer size 250 mm × 130 mm × 45 mm) and placed in a standard curing laboratory for curing.(2)The specimens were cured for 18 h in a curing room at 25 ± 1 °C with relative humidity 75 ± 5% before demolding.(3)When maintenance began, the strain gauge was pasted to collect data and observe the initial time of crack emergence. When the first crack was found, artificial observations were performed every 15 min for 12 h, then once a day afterwards.(4)The crack width of the elliptical ring of foamed concrete was observed with a 100-fold reading microscope, with a light source, and the development of the maximum crack width was observed and recorded over time.(5)The crack width was recorded with a reading microscope. The crack width was the average value of the crack width at 1/4, 1/2 and 3/4 of the height direction of the elliptical ring sample. The time of crack initiation, and the number and width of cracks in eadddch foamed concrete ellipse specimen were also observed and recorded. The test specimen is shown in Figure 5.

## 3. Results

### 3.1. Effect of Different Curing Regimes on Drying Shrinkage of Foamed Concrete

#### 3.1.1. Drying Shrinkage Test Data of Foamed Concrete

The shrinkage data for foamed concrete during the curing regime were recorded using a comparator, as listed in Table 4.

#### 3.1.2. Variance Analysis of the Influence of Various Factors on the Drying Shrinkage of Foamed Concrete

The test data were analyzed by multi-factor analysis of variance, with mathematical statistical software SPSS (Statistical Product and Service Solutions) [23]; the results are shown in Table 5.

According to Table 5, among the three factors that affect the shrinkage of foamed concrete, the sum of the mean deviation of temperature is 58,129.167, which is larger than the other factors. Meanwhile, the corresponding F value is also the largest, and the significance (Sig) is the smallest. It can be seen that temperature has the strongest influence on the drying shrinkage of foamed concrete, followed by water–cement ratio, and finally humidity. From the pairwise interaction analysis of these three factors, it can be seen that the Sig value of humidity × water–cement ratio is greater than 0.05, so the pairwise interaction of the three factors is not obvious.

As can be seen from Table 6, the estimated marginal mean value of shrinkage decreases successively as the temperature increases from 25 °C to 65 °C, indicating that the shrinkage of foamed concrete becomes higher as the temperature rises. It can be seen from Table 7 that 75% relative humidity has little influence on the shrinkage of foamed concrete. It can be concluded from Table 8 that the shrinkage of foamed concrete is slighter when the water–cement ratio is 0.4. In summary:(1)When the humidity and water–cement ratio are the same, the 28-day shrinkage of foamed concrete increases with the increase in temperature;(2)When the humidity and temperature are the same, the 28-day shrinkage of foamed concrete decreases with the increase in water–cement ratio;(3)Temperature is the primary factor that affects the drying shrinkage performance of foamed concrete, followed by water–cement ratio and humidity.

#### 3.1.3. Effect of Temperature on Drying Shrinkage of Foamed Concrete

Figure 6 shows the drying shrinkage test results of foamed concrete in each group, with a curing condition of 75% relative humidity.

Figure 7 shows the drying shrinkage test results of foamed concrete in each group, with a curing condition of 15% relative humidity.

As the temperature rises, the shrinkage of the foamed concrete continues to increase. High temperature changes the drying shrinkage development process of foamed concrete. The drying shrinkage rate of foamed concrete increases rapidly in the early stage, and it then becomes stable afterward. In the initial stage, generally within the first 5 days, the shrinkage rate of foamed concrete is the largest, and with the extension of curing age, the shrinkage rate gradually slows down. When RH = 75%, the drying shrinkage of foamed concrete with different water–cement ratios tends to be the same, while when RH = 15%, the drying shrinkage is obviously different, due to different water–cement ratios. The shrinkage of foamed concrete obviously decreases with the decrease in water–cement ratio in a dry environment. At the same time, with the increase in water–cement ratio, the total shrinkage deformation of foamed concrete increases at 28 days.

#### 3.1.4. Effect of Humidity on Drying Shrinkage of Foamed Concrete

Figure 8 shows the drying shrinkage test results of foamed concrete in different humidity at 25 °C and 65 °C.

As can be seen in Figure 7, the shrinkage of foamed concrete is significantly greater with low humidity. Increasing the humidity in the curing environment can control the loss rate of water in foamed concrete, thus reducing the drying shrinkage value and effectively improving the volume stability of foamed concrete. As the temperature rises, the difference in drying shrinkage under humidity of 15% and 75% is reduced. When the temperature is 25 °C, the shrinkage difference between RH15% and RH75% is 96 × 10^−6^. When the temperature is 65 °C, the shrinkage difference of foamed concrete under different humidity is 66 × 10^−6^.

### 3.2. Influence of Different Curing Regimes on Early Cracking of Foamed Concrete

#### 3.2.1. Early Cracking Test Data of Foamed Concrete

Table 9 shows the initial cracking time of foamed concrete in six different curing systems.

#### 3.2.2. Effect of Temperature and Humidity on Early Cracking of Foamed Concrete

Figure 9 reflects the influence of different curing conditions on the initial cracking time of foamed concrete. It can be seen from the figure that the initial cracking time of foamed concrete decreases with the increase in temperature. When the temperature is 25 °C, the maximum initial cracking time of foamed concrete is 70.8 h. The cracking sensitivity of foamed concrete increases as temperature increases. Combined with the drying shrinkage analysis of foamed concrete, it can be seen that with the increase in temperature, the early hydration speed of foamed concrete increases, and the shrinkage increases, which accelerates the appearance of cracks. Humidity has a great influence on the initial cracking time of foamed concrete. In the same temperature conditions, the difference in initial cracking time caused by different humidity is up to 32 h. The initial cracking time of foamed concrete decreases with the decrease in humidity.

#### 3.2.3. Effect of Water–Cement Ratio on Early Cracking of Foamed Concrete

Figure 10 shows the early cracking of foamed concrete when the water–cement ratios are 0.4 and 0.5.

As shown in Figure 10, different water–cement ratios lead to changes in initial cracking times. The initial cracking time of a specimen with a water–cement ratio of 0.4 is longer. When the water–cement ratio increases to 0.5, the initial cracking time is shortened by 1–5 h. This shows that under the same environmental conditions, the water–cement ratio has a greater influence on the cracking of foamed concrete, i.e., with the increase in water–cement ratio, the cracking of foamed concrete increases.

#### 3.2.4. Effect of Temperature and Humidity on Maximum Crack Width of Foamed Concrete

As shown in Figure 11, with the increase in temperature, the maximum crack width of foamed concrete presents an increasing trend. At relative humidity of 75%, the maximum crack width increases linearly from 1.1 mm to 2.7 mm when the temperature increases from 25 to 65 °C. The crack width expands further when the ambient humidity is lowered to 15%. At the temperature of 65 °C and relative humidity of 15%, the maximum crack width reaches 3.3 mm.

## 4. Discussion

In the case study of the China–Russian Eastern Gas Pipeline, the shrinkage and cracking of foamed concrete was studied under typical environments, where temperature, humidity and water–cement ratio were representative of project scenarios. The results show that the decrease in humidity and the increase in temperature will reduce the shrinkage and cracking of foamed concrete. During the test, it was found that the range of water–cement ratio 0.4–0.5 can meet the shrinkage and cracking requirements of foamed concrete. In simulated conditions, the maximum crack width of foamed concrete was 3.3 mm, which meets the requirements of the “*Technical Regulations for Filling Lightweight Foamed Soil Rail Transit*” (CECS453-2016) [24]. The width of the unstressed penetration crack should be less than 5 mm. Through the analysis of variance, the interaction between the three factors was studied, and the results show that: in the two-by-two interaction analysis between the three factors, the least significant is the Sig value of humidity × water–cement ratio greater than 0.05; the pairwise interaction of the three factors is, therefore, not obvious. In addition, the results show that a combination of temperatures of 25 ± 1 °C and RH 75 ± 5% has the best effect on inhibiting the shrinkage and cracking of foamed concrete. This provides a reference for the maintenance of large-volume foamed concrete, but there are many other factors [25] that need to be considered in order to obtain the best performance, such as apparent density [26] and proportions of concrete mix. Therefore, a large number of relevant tests are needed.

## Figures and Tables

**Figure 1 polymers-14-02703-f001:**
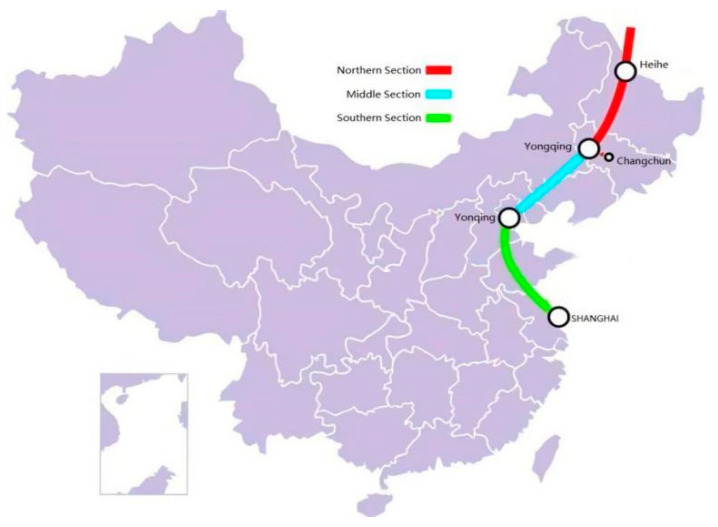
The route of the East China–Russia pipeline.

**Figure 2 polymers-14-02703-f002:**
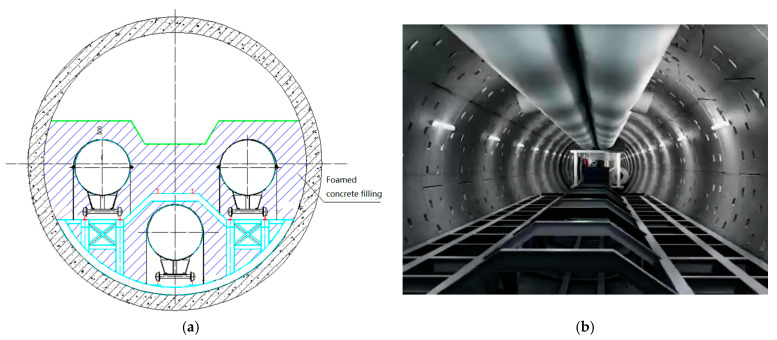
Schematic diagram of foamed concrete tunnel filling: (**a**) sectional structure diagram; (**b**) diagram of the tunnel interior.

**Figure 3 polymers-14-02703-f003:**
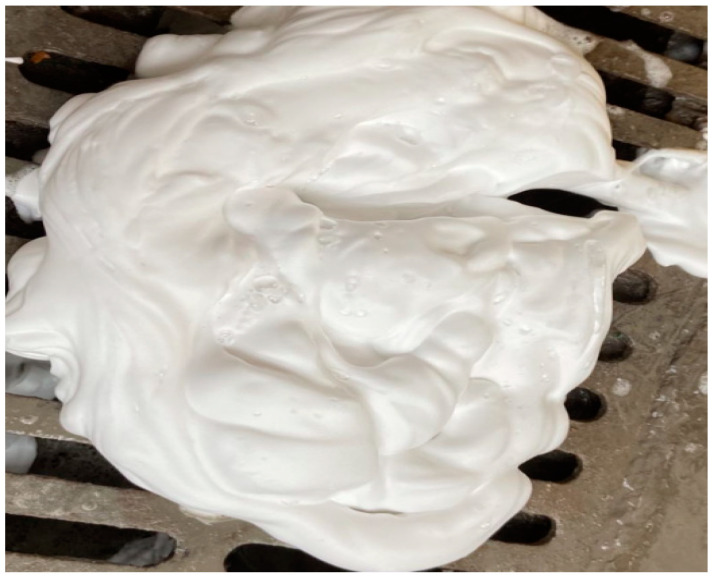
Foaming agent for HT foamed concrete.

**Figure 4 polymers-14-02703-f004:**
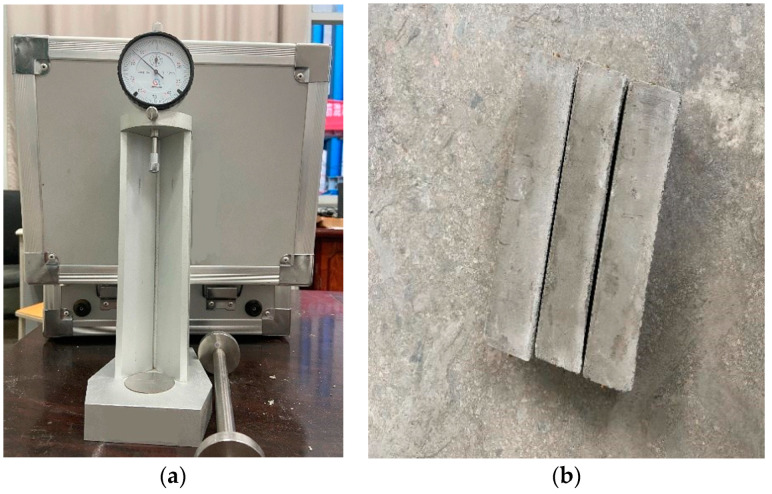
Drying shrinkage test: (**a**) comparator; (**b**) drying shrinkage specimen.

**Figure 5 polymers-14-02703-f005:**
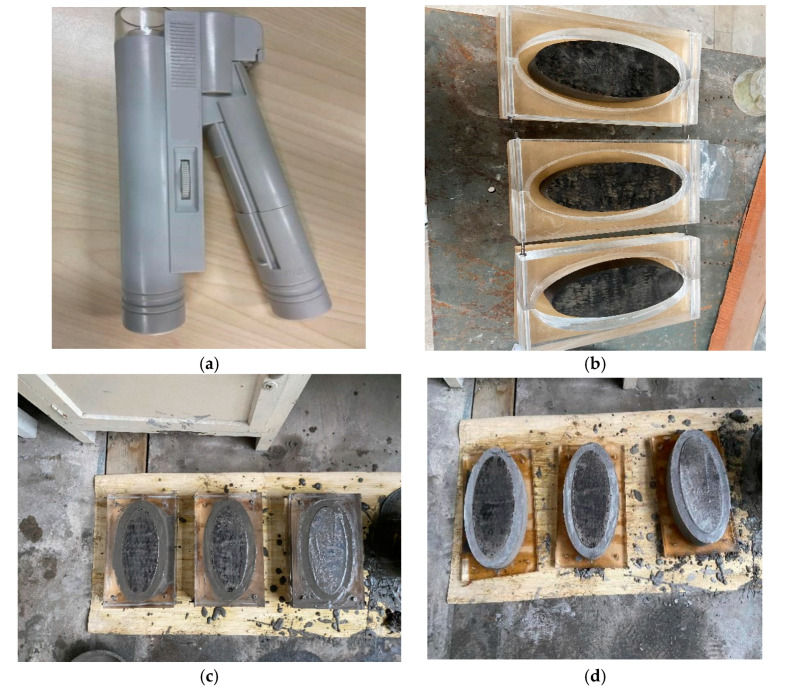
Apparatus for cracking test: (**a**) reading microscope; (**b**) oval mold; (**c**) casting the specimens; (**d**) specimen forming.

**Figure 6 polymers-14-02703-f006:**
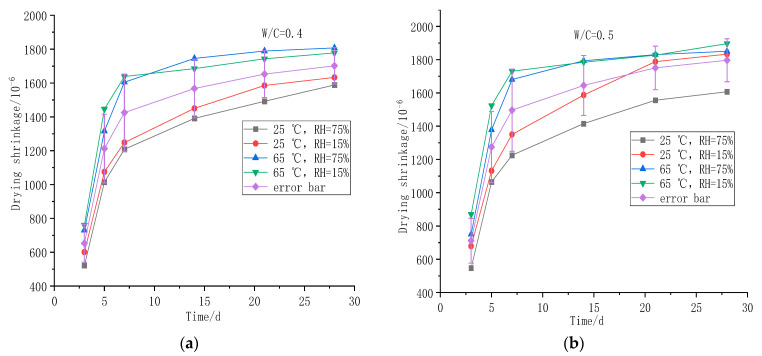
Drying shrinkage of foamed concrete at different curing temperatures: (**a**) 0.4 water–cement ratio dry shrinkage of foamed concrete; (**b**) 0.5 water–cement ratio dry shrinkage of foamed concrete.

**Figure 7 polymers-14-02703-f007:**
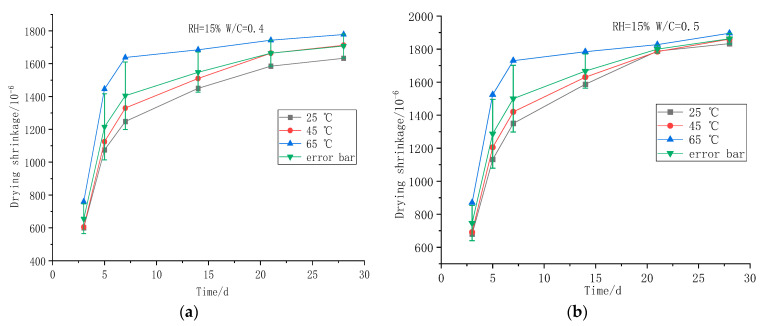
Drying shrinkage of foamed concrete under different curing temperatures (RH = 15%): (**a**) 0.4 water–cement ratio dry shrinkage of foamed concrete; (**b**) 0.5 water–cement ratio dry shrinkage of foamed concrete.

**Figure 8 polymers-14-02703-f008:**
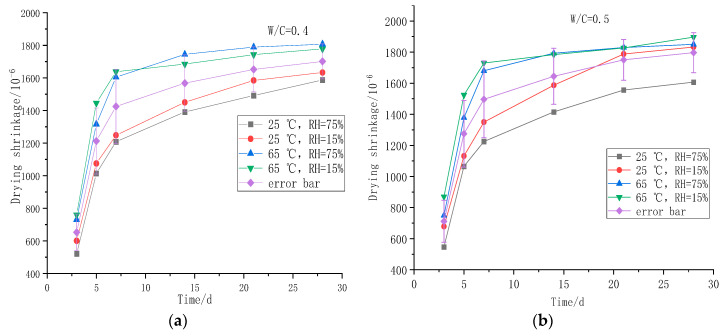
Drying shrinkage of foamed concrete under different curing humidity: (**a**) 0.4 water–cement ratio dry shrinkage of foamed concrete; (**b**) 0.5 water–cement ratio dry shrinkage of foamed concrete.

**Figure 9 polymers-14-02703-f009:**
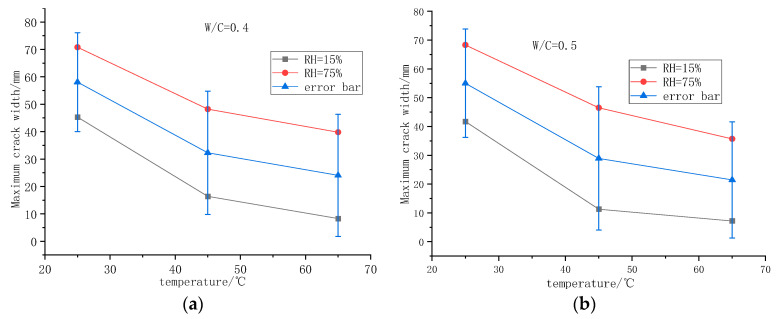
Initial cracking time of foamed concrete under different curing conditions: (**a**) 0.4 water–cement ratio initial cracking time of foamed concrete; (**b**) 0.5 water–cement ratio initial cracking time of foamed concrete.

**Figure 10 polymers-14-02703-f010:**
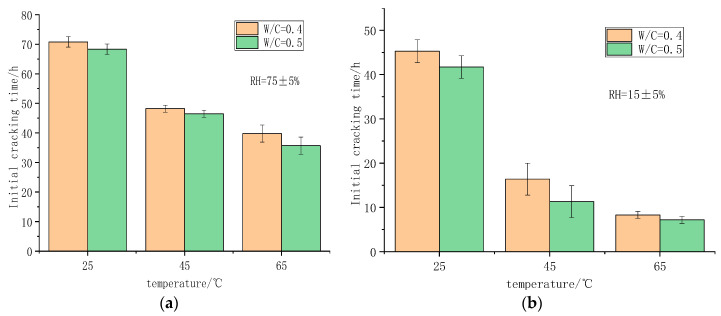
Initial cracking time of foamed concrete with different water–cement ratios: (**a**) RH = 75% initial cracking time of foamed concrete; (**b**) RH = 15% initial cracking time of foamed concrete.

**Figure 11 polymers-14-02703-f011:**
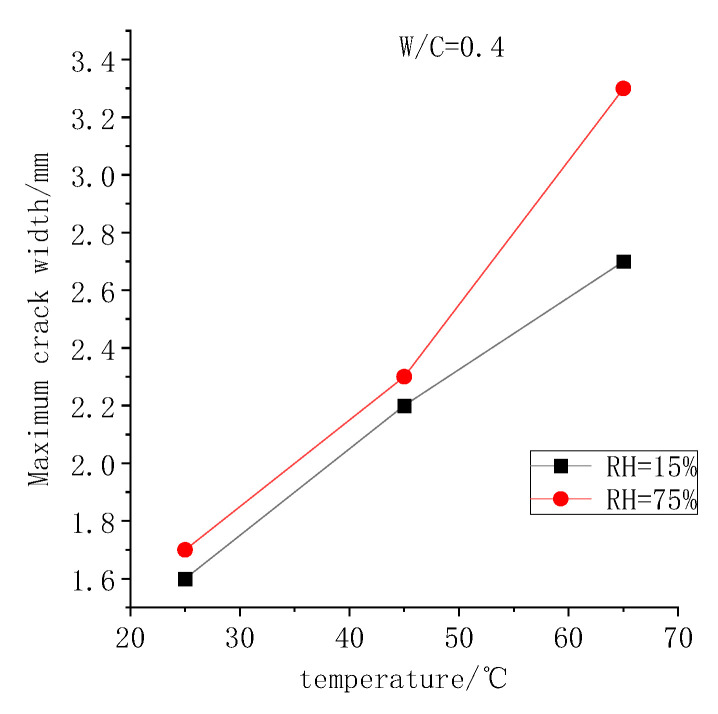
Maximum crack width of foamed concrete under different curing environments.

**Table 1 polymers-14-02703-t001:** Cement physical performance indicators.

Label	Initial Setting Time/min	Final Setting Time/min	Packing Density/Kg/m^3^	The Compressive Strength/Mpa			Flexural Strength/MPa		
				3 Day	7 Day	28 Day	3 Day	7 Day	28 Day
**Ordinary Portland cement**	70	242	1137	18.5	28	47	3.6	5.1	7.3

**Table 2 polymers-14-02703-t002:** Chemical composition of fly ash.

Composition	SO_3_	MgO	CaO	Fe_2_O_3_	Al_2_O_3_	SiO_2_	Ignition Loss
**Content**	0.66%	1.09%	3.53%	4.04%	30.37%	50.76%	3.32%

**Table 3 polymers-14-02703-t003:** Foamed concrete base mix ratio.

Number	Water-Cement Ratio	The Fly Ash/kg	Cement/kg	Water/kg	HTFC Mineral Admixtures/kg
1	0.4	202	552	368	166
2	0.5	202	552	460	166

**Table 4 polymers-14-02703-t004:** Dry shrinkage of foamed concrete in different environments.

Number	Water-Cement Ratio	Maintenance Time before Mold Removal	The Shrinkage of the Specimen (×10^−6^)					
			3 d	5 d	7 d	14 d	21 d	28 d
1	0.4	1 d	521	1014	1208	1391	1492	1588
		3 d	394	933	1135	1321	1410	1480
	0.5	1 d	547	1065	1225	1415	1556	1607
		3 d	472	984	1193	1370	1484	1507
2	0.4	1 d	601	1075	1248	1450	1585	1633
		3 d	580	990	1195	1387	1465	1580
	0.5	1 d	679	1132	1350	1587	1788	1833
		3 d	620	1078	1355	1468	1583	1654
3	0.4	1 d	624	1080	1325	1543	1680	1740
		3 d	605	999	1200	1407	1545	1617
	0.5	1 d	650	1195	1380	1576	1725	1794
		3 d	590	1026	1214	1423	1550	1655
4	0.4	1 d	605	1125	1330	1510	1664	1712
		3 d	609	1035	1270	1478	1624	1695
	0.5	1 d	691	1205	1420	1630	1785	1860
		3 d	658	1130	1325	1505	1695	1740
5	0.4	1 d	730	1316	1605	1745	1789	1807
		3 d	720	1280	1410	1590	1695	1780
	0.5	1 d	750	1378	1680	1793	1830	1850
		3 d	727	1240	1420	1579	1680	1726
6	0.4	1 d	760	1446	1638	1685	1743	1778
		3 d	705	1403	1587	1615	1874	1740
	0.5	1 d	870	1524	1730	1784	1827	1896
		3 d	820	1480	1670	1706	1801	1854

**Table 5 polymers-14-02703-t005:** Analysis of the effect of various factors on the shrinkage of 28-day foamed concrete.

The Source of the Variance	The Sum of the Deviations from the Mean	Degree of Freedom (df)	Average Variance	F	Sig
**Temperature**	58,129.167	1	29,064.583	36.395	0.027
**Humidity**	28,856.33	1	8856.33	31.090	0.030
**Water–Cement Ratio**	28,227.000	1	28,227.000	35.346	0.027
**Temperature** **×** **Humidity**	9937.167	2	4960.583	6.2222	0.138
**Humidity** **×** **Water–Cement Ratio**	10,208.333	1	10,208.333	12.783	0.070
**Temperature** **×** **Water–Cement Ratio**	444.500	2	222.250	0.278	0.782

**Table 6 polymers-14-02703-t006:** Analysis of the effect of temperature on the shrinkage of 28-day foamed concrete.

Temperature	Average Value	Estimate the Marginal Mean	95%Confidence Interval	
			Lower Limit	Upper Limit
25	1665.3	14.1	1604.5	1726.1
45	1776.5	13.8	1715.7	1837.3
65	1832.7	13.7	1771.9	1893.5

**Table 7 polymers-14-02703-t007:** Analysis of the effect of humidity factors on the shrinkage of 28-day foamed concrete.

Humidity	Average Value	Estimated Marginal Value	95% Confidence Interval	
			Lower Limit	Upper Limit
15	1785.3	10.5	1735.7	1834.9
75	1735.0	11.5	1681.3	1780.7

**Table 8 polymers-14-02703-t008:** Analysis of the effect of water–ash ratio factors on the shrinkage of 28-day foamed concrete.

Water-Cement Ratio	Average Value	Estimated Marginal Value	95% Confidence Interval	
			Lower Limit	Upper Limit
0.4	1709.7	11.5	1660.0	1759.3
0.5	1806.7	10.5	1757.0	1856.3

**Table 9 polymers-14-02703-t009:** Initial cracking time under different curing environments (unit in the table: h).

	Water-Cement Ratio	0.4	0.5
Number			
1		70.8	68.3
2		45.3	41.7
3		48.2	46.5
4		16.4	11.3
5		39.8	35.7
6		8.3	7.2

## Data Availability

The data used to support the findings of this study are available from the corresponding author upon request.

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
