# Peer review of "Effect of Maintenance and Water–Cement Ratio on Foamed Concrete Shrinkage Cracking"

_polymers, 2022, doi:10.3390/polym14132703_

Round 1
Reviewer 1 Report
Very interesting topic! Authors managed to study and characterize the foamed concrete expansively, and their claims are backed up by experimental results.
Line 36 to 44, and 298-301 why did you not delete the template sentences? I was shocked at first as I couldn't figure out the relationship between those words and your research? You need to ask someone to check the manuscript before submission, I do have a feeling this paper was not checked by other authors!!
Chinese standards are not common outside China, please give reference to them so we can find them online and check the settings. As an example: JGJ/T 341-2014
Figure 4, 5 the picture on the right is distorted, also you need to label each individual picture and describe them properly in the Figure caption.
Figure 6 to 11 how many repetitions did you make from each sample category? You need to use error bars and mention the number of repetitions.
Reviewer 2 Report
The Authors of the work presented interesting research on shrinkage cracking of foamed concrete. The influence of the setting temperature, humidity and water-cement ratio on shrinkage and cracking resistance of foamed concrete was investigated.
The work requires minor corrections:
The Authors of the work did not show due diligence when it comes to preparing the text of the work. Using the template prepared by the journal in the text of the paper, they did not remove the guidelines contained in the template.
For example: In the introduction, lines 36 to 44 should be deleted as completely unrelated to work.
Similarly, in point 4 of Discussion - lines 298 to 301 should be deleted.
In table 9 it should be stated that the numbers in the table represent hours.
Round 2
Reviewer 1 Report
approved